# The targetable kinase PIM1 drives ALK inhibitor resistance in high-risk neuroblastoma independent of MYCN status

Ricky M. Trigg [1,9,12], Liam C. Lee[1,10,12], Nina Prokoph[1,12], Leila Jahangiri[1], C. Patrick Reynolds[2], G.A. Amos Burke[3], Nicola A. Probst[1], Miaojun Han[1,11], Jamie D. Matthews[1], Hong Kai Lim[1], Eleanor Manners[1], Sonia Martinez [4], Joaquin Pastor[4], Carmen Blanco-Aparicio [4], Olaf Merkel[5], Ines Garces de los Fayos Alonso[5], Petra Kodajova[6], Simone Tangermann[6], Sandra Högler[6], Ji Luo[7], Lukas Kenner [5,6,8,12] & Suzanne D. Turner [1,12]*

Resistance to anaplastic lymphoma kinase (ALK)-targeted therapy in ALK-positive non-small cell lung cancer has been reported, with the majority of acquired resistance mechanisms relying on bypass signaling. To proactively identify resistance mechanisms in ALK-positive neuroblastoma (NB), we herein employ genome-wide CRISPR activation screens of NB cell lines treated with brigatinib or ceritinib, identifying *PIM1* as a putative resistance gene, whose high expression is associated with high-risk disease and poor survival. Knockdown of *PIM1* sensitizes cells of differing *MYCN* status to ALK inhibitors, and in patient-derived xenografts of high-risk NB harboring ALK mutations, the combination of the ALK inhibitor ceritinib and PIM1 inhibitor AZD1208 shows significantly enhanced anti-tumor efficacy relative to single agents. These data confirm that *PIM1* overexpression decreases sensitivity to ALK inhibitors in NB, and suggests that combined front-line inhibition of ALK and PIM1 is a viable strategy for the treatment of ALK-positive NB independent of *MYCN* status.

[1] Division of Cellular and Molecular Pathology, Department of Pathology, University of Cambridge, Lab Block level 3, Box 231, Cambridge Biomedical Campus, Cambridge CB2 0QQ, UK. [2] Cancer Center, Texas Tech University Health Sciences Center School of Medicine, Lubbock, TX 79430, USA. [3] Department of Paediatric Oncology, Box 181, Cambridge University Hospitals NHS Foundation Trust, Cambridge Biomedical Campus, Hills Road, Cambridge CB2 0QQ, UK. [4] Experimental Therapeutics Programme, Spanish National Cancer Research Centre (CNIO), Madrid, Spain. [5] Department of Experimental Pathology and Laboratory Animal Pathology, Institute of Clinical Pathology, Medical University of Vienna, Währinger Gürtel 18-20, Vienna 1090, Austria. [6] Unit of Laboratory Animal Pathology, University of Veterinary Medicine Vienna, Veterinärplatz 1, Vienna 1210, Austria. [7] Laboratory of Cancer Biology and Genetics, Center for Cancer Research, National Cancer Institute, National Institutes of Health, Bethesda, MD 20814, USA. [8] Christian Doppler Laboratory for Applied Metabolomics (CDL-AM), Boltzmanngasse 20, Medical University of Vienna, Vienna 1090, Austria. [9] Present address: Functional Genomics, Medicinal Science & Technology, GlaxoSmithKline, Stevenage SG1 2NY, UK. [10] Present address: Amgen, Thousand Oaks, CA 91320, USA. [11] Present address: OncoSec, San Diego, CA 92121, USA. [12] These authors contributed equally: Ricky M. Trigg, Liam C. Lee, Nina Prokoph, Lukas Kenner, Suzanne D. Turner. *email: sdt36@cam.ac.uk

Deriving from precursor cells of the sympathetic nervous system, neuroblastoma (NB) is the most common and deadly extracranial solid tumor in children[1,2]. NB presents at various sites along the sympathoadrenal axis, most commonly in the adrenal medulla or paraspinal ganglia[3]. Characterized by heterogeneous biological and clinical features ranging from spontaneous regression to aggressive treatment-resistant disease, NB is often referred to as a clinical enigma. While low- and intermediate-risk forms of NB are highly curable, over half of patients with high-risk disease suffer relapse and five-year survival is 40–50%[4]. Therefore, novel treatment strategies aimed at providing long-term disease remission are urgently sought.

Large-scale genomic studies are bringing the genetic basis of NB into focus, and there is evidence to suggest that high and low-risk forms of the disease evolve through distinct genetic mechanisms[5]. A study of 493 NB cases employing comparative genomic hybridization showed hyperdiploidy involving whole-chromosome gains to be associated with low-risk NB, while segmental chromosome aberrations were associated with high-risk NB[6]. Recent genome-wide sequencing analyses in large NB patient cohorts have identified a relative paucity of recurrent somatic mutations[7–9].

Anaplastic lymphoma kinase (ALK) is the most commonly mutated gene in NB, where gain-of-function mutations in the kinase domain – namely at residues F1174, F1245, and R1275 – are found in 8–10% of cases overall[10,11]. An additional 2–3% of patients harbor focal amplification of *ALK*, and this feature correlates with poor survival[10–12]. Given the plethora of interest in the development of ALK inhibitors in non-small cell lung cancer (NSCLC), the assessment of these compounds in ALK-positive NB quickly followed. Numerous recent studies have demonstrated the efficacy of ALK inhibitors against ALK-positive NB cell lines and patient-derived xenografts[13–15]. Several of these studies have documented the de novo resistance of the ALK$^{F1174L}$ mutation to crizotinib and ceritinib, and have devised combinatorial treatment strategies to enhance efficacy[13,16–19].

In patients with ALK-positive NSCLC, acquired resistance has been shown to arise with first, second and third-generation ALK inhibitors, presenting a major challenge in the long-term use of these compounds[20]. The most common mechanisms of resistance to ALK inhibition in NSCLC are reported to involve bypass signaling through functionally-related pathways, such that cells grow in an ALK-independent manner when the kinase is inhibited[21]. Bypass mechanisms include activation of EGFR[22], HER2[23], KRAS[24], and IGF-1R[25], and amplification of *KIT*[22].

To identify mechanisms of resistance to ALK inhibitors in ALK-positive NB that involve bypass signaling, we conduct genome-wide CRISPR activation (CRISPRa) screens[26] in the NB cell lines SH-SY5Y (ALK$^{F1174L}$) and CHLA-20 (ALK$^{R1275Q}$) under treatment with brigatinib or ceritinib for 14 days. We identify 25 putative resistance genes in SH-SY5Y, and successfully validate 21 genes in vitro, and we further characterize *PIM1* given its association with high-risk disease and poor survival outcomes in NB. Indeed, overexpression or knockdown of *PIM1* induces resistance or sensitization to ALK inhibitors, respectively, and combinations of ALK inhibitors with AZD1208, a small-molecule pan-PIM inhibitor, demonstrate at least additive effects if not mild-to-moderate synergy in vitro. Moreover, in patient-derived xenograft (PDX) models of high-risk NB harboring ALK$^{F1245C}$ or ALK$^{F1174L}$, the antitumor efficacy of ceritinib and AZD1208 is significantly greater than either agent alone. Finally, overexpression of *PIM1* is similarly found to induce resistance to brigatinib and ceritinib in cell lines derived from ALK-positive anaplastic large cell lymphoma (ALCL). These data implicate *PIM1* in ALK inhibitor resistance in ALK-positive NB and other ALK-driven malignancies, suggesting that combined pharmacological inhibition of ALK and PIM1 may be beneficial in the treatment of ALK-positive, high-risk NB.

## Results

**CRISPRa screens identify ALK inhibitor resistance genes.** Prior to CRISPRa screens, the NB cell lines SH-SY5Y (ALK$^{F1174L}$) and CHLA-20 (ALK$^{R1275Q}$) were characterized for their sensitivity to ALK tyrosine kinase inhibitors (TKIs) in order to determine the ED$_{50}$ and ED$_{75}$ concentrations (Supplementary Table 1, Supplementary Fig. 1a–c). Cells were then transduced with lentiviral constructs to express the CRISPR-based synergistic activation mediator (SAM) complex[26]. The functionality of the complex was validated by transducing cells with gRNAs specific to 15 genes previously shown to confer resistance to ALK inhibition in NSCLC[23]. These data showed significant overexpression of 6/15 and 8/15 genes in SH-SY5Y and CHLA-20 cell lines, respectively, although a third of genes were not significantly overexpressed in either cell line (Supplementary Fig. 1d).

The SAM pooled gRNA library, targeting the transcription start site of 23,430 RefSeq coding isoforms with three gRNA sequences, was used for CRISPRa screening[26]. Cells were transduced with the gRNA library before exposure to either brigatinib or ceritinib. Genomic DNA was extracted from cells at days 0 and 14, and deep-sequencing conducted to identify enriched gRNAs (Fig. 1a). To further increase the stringency of the analysis, we considered candidate genes to be those with enriched gRNAs when exposed to both brigatinib or ceritinib at a given concentration (i.e., ED$_{50}$ or ED$_{75}$) (Supplementary Data 1).

For SH-SY5Y cells, ALK TKI inhibitor treatment led to significant enrichment of gRNAs targeting 25 genes ($p < 0.05$, unpaired Student's t-test) (Fig. 1b). Interestingly, only three genes (*PIM1*, *MET* and *SAGE1*) were significantly enriched by both inhibitors at ED$_{50}$ and ED$_{75}$ concentrations (Fig. 1b–d), suggesting resistance mechanisms may be compound-specific and concentration-dependent. Gene set enrichment analysis of candidate resistance genes in SH-SY5Y cells in comparison with hallmark and gene ontology gene sets in the Molecular Signatures Database[27] showed an enrichment for genes involved in negative regulation of cell death (*PIM1*, *BDNF*, *FAIM2*, *FOXP1*, *KRAS*, *MET*, and *MYC*) (Fig. 1e). This is consistent with their putative roles in drug resistance and therefore confirms the predictive capability of this CRISPR gRNA fold-enrichment analysis method. Data obtained from the SH-SY5Y cell line were compared to those from the CRISPRa screen conducted in the CHLA-20 cell line and revealed an overlap of 11 genes including *PIM1*, *MET*, *MYC*, and *KRAS* (Fig. 1f, Supplementary Fig. 2, Supplementary Data 2).

**Functional validation of CRISPRa screen hits.** Candidate genes in SH-SY5Y and CHLA-20 cells were each functionally validated by transducing cells with two enriched gRNAs individually and by assessing their response to brigatinib or ceritinib. Of the gRNAs targeting 25 different genes, 76% (38/50) induced a significant increase in the ED$_{50}$ concentration for both brigatinib and ceritinib in SH-SY5Y cells ($p < 0.05$, one-way ANOVA) (Supplementary Fig. 3a, b; Supplementary Table 2), and 24 genes were validated. Levels of gene overexpression were assessed for all candidate genes by RT-qPCR (Supplementary Fig. 3c). Similar data were obtained for the CHLA-20 cell line (Supplementary Fig. 4). While overexpression levels achieved ranged from ~1.3 to >100,000-fold for both SH-SY5Y and CHLA-20 cell lines, the ability of modest increases in gene expression to induce ALK inhibitor resistance, such as those observed for *BDNF*, *KRAS*,

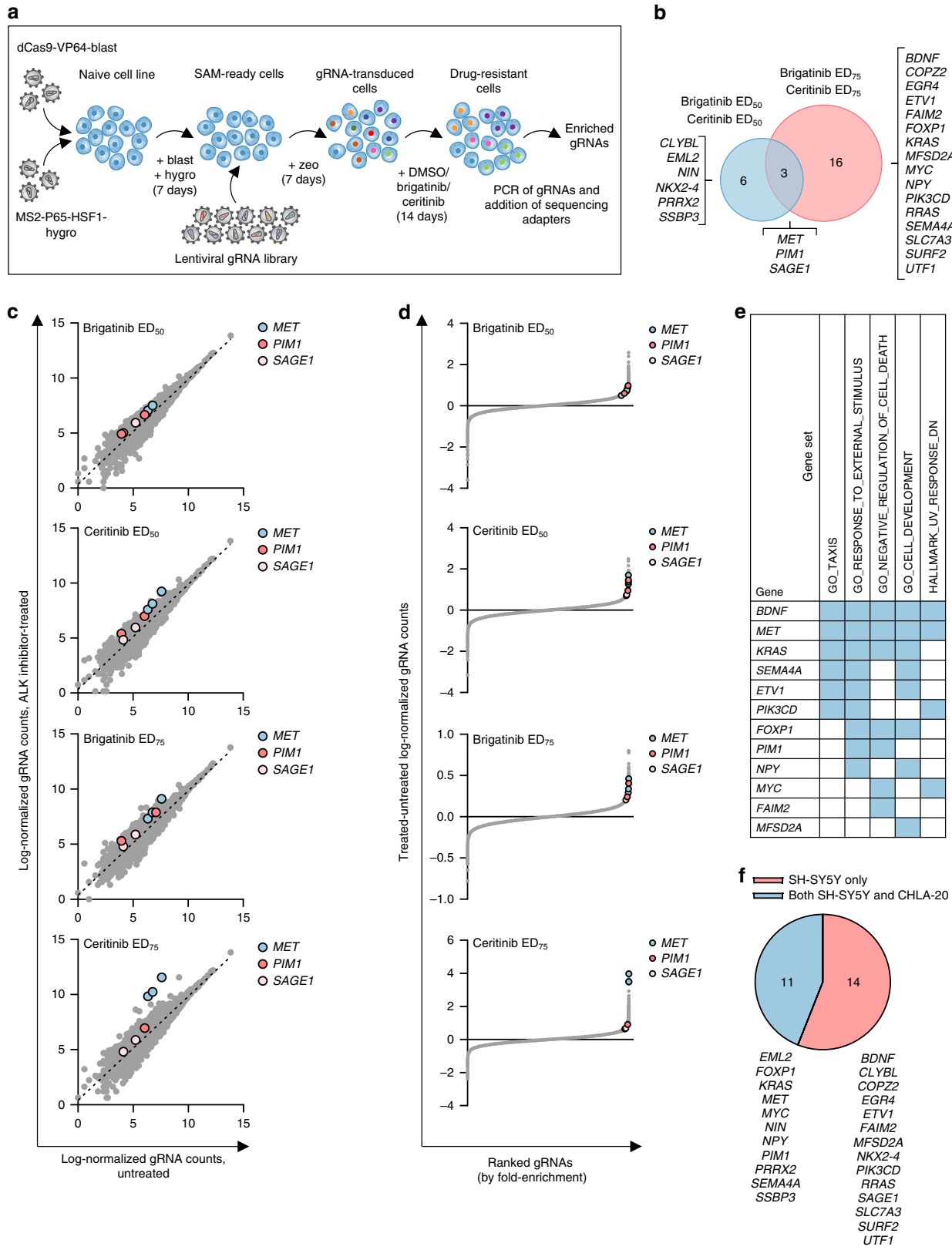

MYC, and RRAS, supports the biological relevance of these results (Supplementary Fig. 3c, Supplementary Fig. 4c).

**PIM1 is a potential druggable target for ALK TKI-resistant NB.** To identify clinically-relevant genes from the CRISPRa screens with the greatest potential to modify sensitivity to ALK

inhibitors, Kaplan–Meier overall and event-free survival analyses were conducted for each gene ($n = 24$) based on its expression level in primary ALK inhibitor-naive tumors from large NB datasets (all risk groups) published by Zhang and colleagues[28] and Kocak and colleagues[29]. High gene expression was associated with significantly worse overall survival ($p < 0.01$, Bonferroni-

**Fig. 1** A genome-wide CRISPRa screen identifies resistance genes in SH-SY5Y cells. **a** Experimental schema for genome-wide CRISPRa screening in NB cells. Cell lines were transduced with lentiviral vectors to confer stable expression of dCas9-VP64 and MS2-p65-HSF1 before transduction with a lentiviral library of guide RNA (gRNA) sequences (3 gRNAs per coding isoform). Transduced cells were selected and then exposed to DMSO (vehicle) or ALK inhibitors for 14 days, after which genomic DNA was extracted and gRNA sequences were PCR-amplified and subjected to Illumina HiSeq sequencing. **b** Venn diagram of candidate genes from CRISPRa screen with brigatinib or ceritinib at $ED_{50}$ and $ED_{75}$ concentrations in SH-SY5Y cells. **c** Log-normalized read counts for each gRNA in untreated (DMSO) vs ALK TKI-treated SH-SY5Y cell populations. Dashed black lines represent linear regressions. Data represent the average of two biological replicates. Genes common to both brigatinib and ceritinib treatments are shown (*MET*, *PIM1*, *SAGE1*). **d** gRNAs ranked by fold-change vs difference in log-normalized read counts between DMSO and ALK TKI-treated SH-SY5Y populations. Only gene hits common to brigatinib and ceritinib treatments are shown (*MET*, *PIM1* and *SAGE1*). **e** Top five gene sets by gene set enrichment analysis including all validated genes ($n = 25$) identified by the CRISPRa screen in SH-SY5Y cells treated with brigatinib or ceritinib. Validated genes were compared with hallmark and GO gene sets in MSigDB (FDR < 0.0001). Blue shading indicates an enriched gene set. **f** Pie chart to show the overlap in candidate resistance genes detected by the CRISPRa screens in both SH-SY5Y and CHLA-20 cells.

corrected, Log-rank test) and event-free survival ($p < 0.05$, Bonferroni-corrected, Log-rank test) for 42% (10/24) and 46% (11/24) of genes, respectively (Supplementary Table 3).

The potential for therapeutic targeting of each resistance gene with FDA-approved drugs or compounds in clinical development was then assessed using the Drug–Gene Interaction database[30]. Five druggable genes were identified that may be amenable to either direct targeting (*PIM1*, *PIK3CD*, and *MET*) or indirect targeting (*KRAS* and *MYC*). *MET* was the top-ranking resistance gene in the CRISPRa screens, enriched under brigatinib and ceritinib treatments at both $ED_{50}$ and $ED_{75}$ concentrations (Fig. 1, Supplementary Fig. 2). Activation of *MET* has previously been shown to confer resistance to the ALK inhibitor alectinib in NSCLC[31], and resistance can be overcome with crizotinib, a potent inhibitor of MET[32]. Given substantial evidence in the literature that *PIM1* mediates resistance to standard chemotherapy[33–35] as well as molecularly targeted agents[36,37], that high expression of *PIM1* is a poor prognostic indicator in multiple cancers[38–40], and that recently Brunen et al.[41]. identified PIM kinases as potential therapeutic targets in *NF1* wild-type NB, this gene was explored further.

In support of PIM1 as a prognostic biomarker, we found its high expression to be significantly associated with worse overall survival ($p < 0.001$, Bonferroni-corrected, Log-rank test) in an independent cohort of NB patients ($n = 498$) (Supplementary Fig. 5a)[28]. Interestingly, *PIM1* transcript level serves as a prognostic biomarker independent of *MYCN* status in this cohort (Supplementary Fig. 5b–e). However, as reported by Brunen et al.[41], we also found *MYCN* amplification to be a stronger predictor of poor prognosis than *PIM1* (Supplementary Table 4). Moreover, *PIM1* expression is significantly elevated in INSS stage 4 disease relative to earlier stages ($p < 0.005$, unpaired Student's *t*-test) and in patients with high-risk disease ($p < 0.0001$, unpaired Student's *t*-test) (Supplementary Fig. 5e, f)[28].

***PIM1* decreases sensitivity to ALK TKIs via apoptosis evasion**. SH-SY5Y and CHLA-20 cells were transduced with individual *PIM1*- and non-targeting gRNAs, leading to 23- and 14-fold increases in *PIM1* mRNA, respectively (Fig. 2a), and then treated with brigatinib or ceritinib for 72 h (Fig. 2b). Overexpression of *PIM1* induced significant increases in both the brigatinib $ED_{50}$ ($p < 0.005$, unpaired Student's *t*-test) and ceritinib $ED_{50}$ ($p < 0.05$, unpaired Student's *t*-test), thus confirming decreased sensitivity to ALK inhibition in these cell lines. Given that PIM1 has been shown to induce drug resistance in other cancers by evading apoptosis[33,42,43], expression of apoptotic regulators and known substrates downstream of PIM1 were investigated in SH-SY5Y and CHLA-20 cells, following gRNA-induced overexpression of *PIM1*. Specifically, expression of BAD[44] and PRAS40[45] was analysed. An increase in p-BAD (Ser112), but not p-PRAS40, was observed in SH-SY5Y and CHLA-20 cells overexpressing *PIM1*

(Fig. 2c) suggesting that in NB cells, PIM1 induces drug resistance by evading apoptosis through direct phosphorylation of BAD. In further evidence of a survival role for PIM1 in the context of ALK inhibitor treatment, SH-SY5Y cells were treated with 1 μM brigatinib or ceritinib for 48 h and a significantly lower apoptotic cell (annexin V + /PI-) fraction was seen in inhibitor-treated cells overexpressing *PIM1* ($p < 0.0001$, unpaired Student's *t*-test) (Fig. 2d).

***PIM1* decreases sensitivity to ALK TKIs in ALK-positive ALCL**. Given that a strong synergistic effect was previously shown on simultaneous inhibition of ALK and PIM kinases in ALK-positive ALCL cell lines[46], *PIM1* was overexpressed and sensitivity to ALK inhibitors monitored in ALK-positive ALCL cell lines. Karpas 299 and SU-DHL-1 cells were transduced to express components of the CRISPR SAM complex whose activity was confirmed (Supplementary Fig. 6a) before assessing their responses to brigatinib or ceritinib upon overexpression of *PIM1* as conducted previously (Supplementary Fig. 6b, c). Indeed, overexpression of *PIM1* led to drug resistance, evidenced by significant increases in the brigatinib $ED_{50}$ ($p < 0.01$, unpaired Student's *t*-test) and ceritinib $ED_{50}$ ($p < 0.001$, unpaired Student's *t*-test). Therefore, *PIM1* is a potential ALK inhibitor-resistance driver in ALCL and is worthy of further exploration.

**PIM inhibition alone is not efficacious in NB cell lines**. To investigate the potential of PIM1 as a therapeutic target in ALK-positive NB, transcript levels of *PIM1* were assessed by RT-qPCR across 25 NB cell lines that are diverse with respect to *ALK* and *MYCN* status (Fig. 3a). As PIM can exist as one of 3 family members (PIM1, PIM2, and PIM3) with many PIM inhibitors showing overlapping activity, the transcript levels of all 3 proteins was determined in the same 25 cell lines (Supplementary Fig. 7a). For *PIM1*, expression levels varied across the cell lines and there was no strong correlation between expression of the different family members (Supplementary Fig. 7b). The activity of the pan-PIM kinase inhibitor AZD1208, with greatest potency for PIM1, was then determined in ALK-positive NB cell lines by 72-h dose–response and 14-day colony formation assays (Fig. 3b, c). Consistent with data reported by Brunen et al.[41], cell lines were relatively insensitive to AZD1208 at clinically relevant concentrations, with predicted $ED_{50}$ values exceeding 10 μM in 8/8 NB cell lines expressing a range of ALK mutants (Fig. 3c). Similar results were noted in response to treatment with PIMi, another small-molecule pan-PIM kinase inhibitor in preclinical development (Fig. 3c), suggesting that pharmacological inhibition of PIM kinases alone is not a viable therapeutic strategy. The response of ALK-negative NB cell lines to AZD1208 and PIMi was likewise analyzed and a similar response was observed, indicating that the

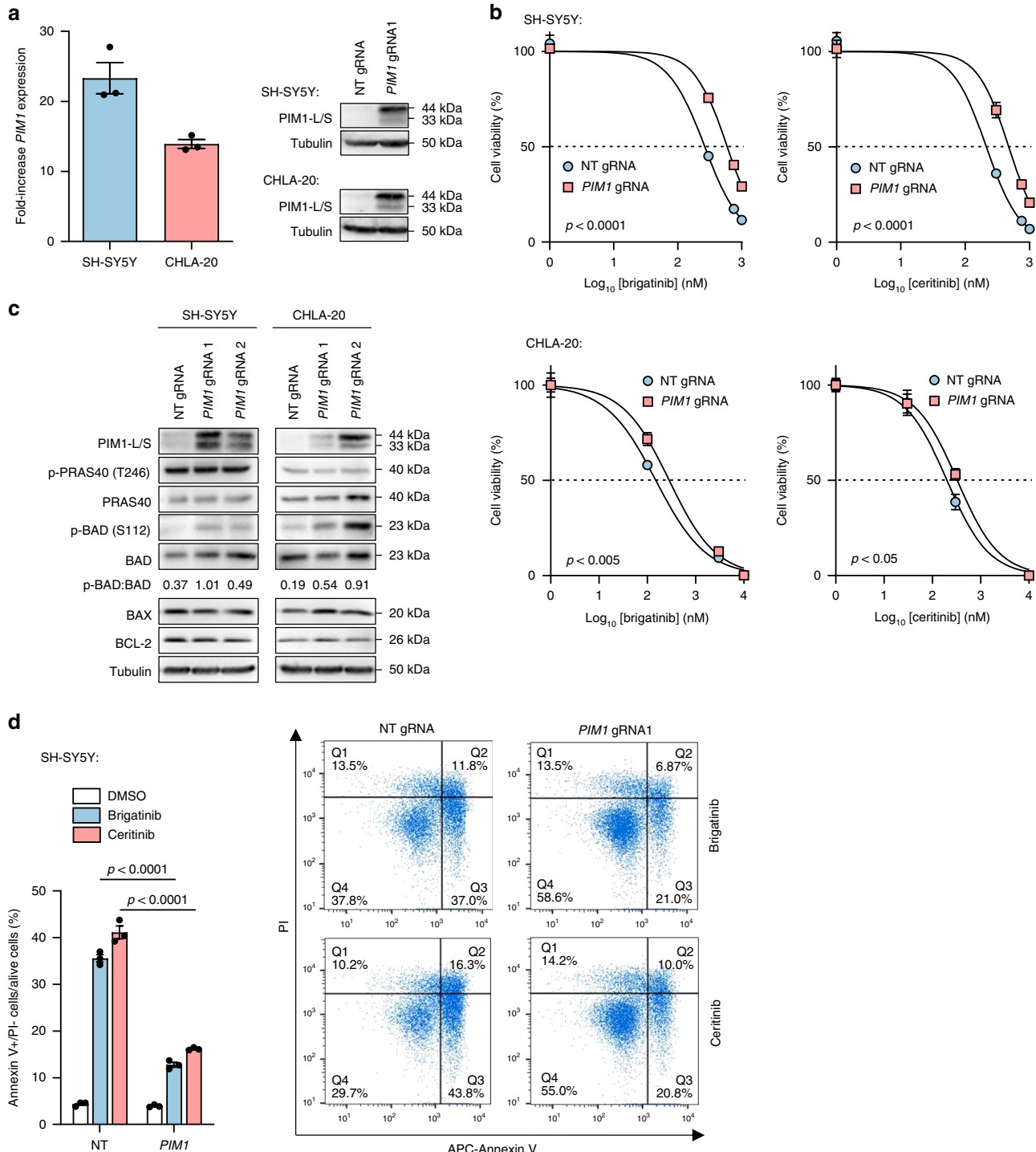

**Fig. 2** *PIM1* inhibits apoptosis through phosphorylation of BAD. **a** Expression of *PIM1* in SH-SY5Y or CHLA-20 cells transduced with non-targeting (NT) or *PIM1*-targeting gRNA molecules, by RT-qPCR and immunoblot. RT-qPCR data were normalized to cells treated with non-targeting gRNA and show means ± s.d. of triplicate experiments. **b** 72-h dose–response assays of brigatinib or ceritinib-exposed SH-SY5Y or CHLA-20 cells transduced with NT or *PIM1*-targeting gRNA. Data points represent means ± s.d. of triplicate experiments. ED$_{50}$ values were compared by an unpaired Student's *t*-test. **c** Immunoblot of known PIM1 targets and apoptotic mediators in SH-SY5Y and CHLA-20 cells transduced with NT- or *PIM1*-targeting gRNA, the latter leading to *PIM1* overexpression. Data are representative of three independent experiments. Numbers below the blot represent the densitometric measurement of pBAD levels normalized to BAD. **d** Percentage of apoptotic (annexin V + /PI-) *PIM1*-overexpressing SH-SY5Y cells after 48-h of treatment with 1 μM brigatinib or ceritinib. Data points in the bar chart show means ± s.d. of triplicate experiments. *p*-values were calculated using an unpaired Student's *t*-test. Source data for this figure are provided as a Source Data file.

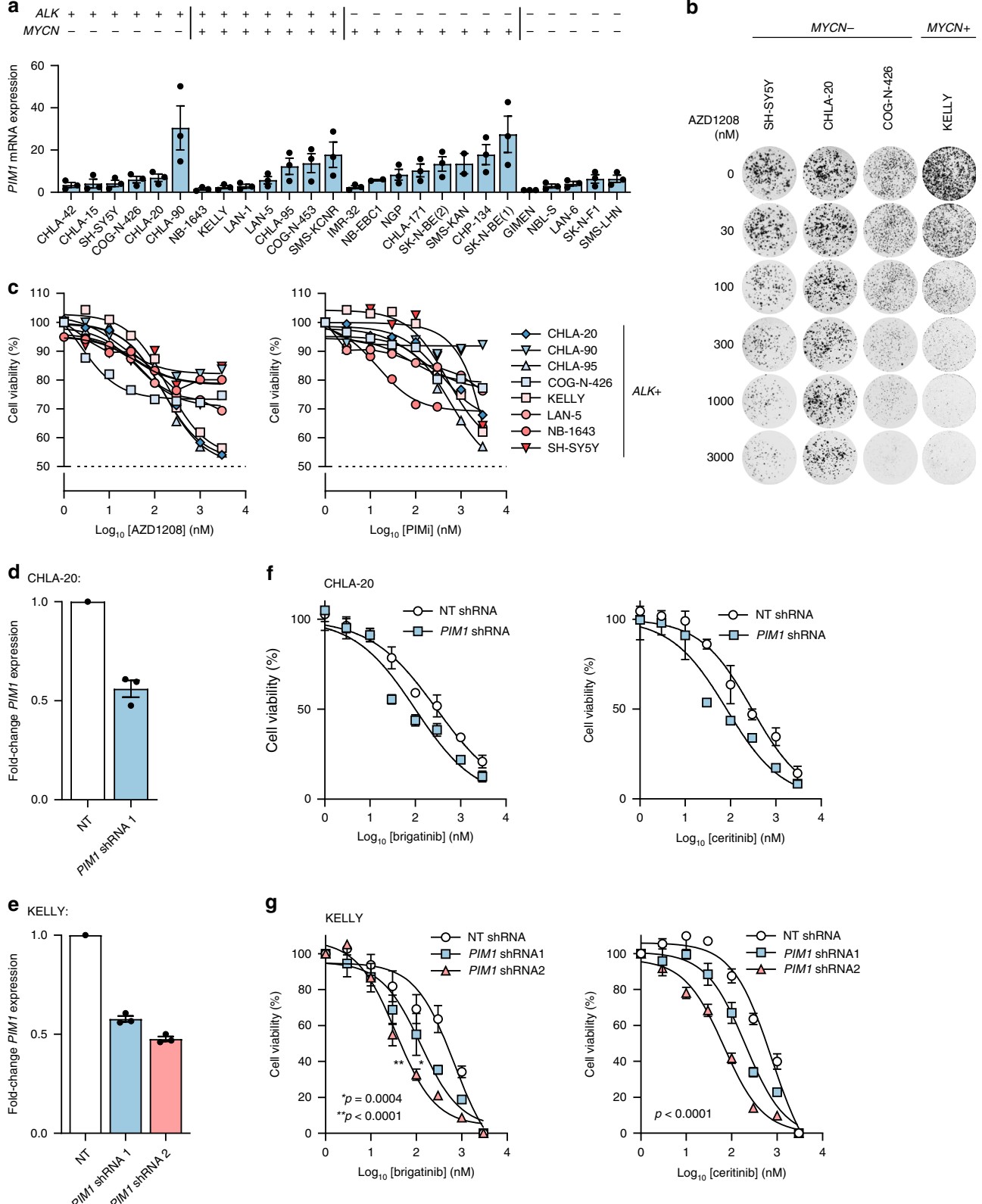

**Fig. 3** PIM1 inhibition lacks potency but enhances the efficacy of ALK inhibitors. **a** RT-qPCR-based expression levels of *PIM1* in 25 neuroblastoma cell lines, normalized to GIMEN cell line expression levels. Data represent means ± s.d. of technical triplicates. **b** 14-day colony formation assay of ALK-positive NB cell lines treated with the indicated doses of AZD1208. **c** 72-h dose–response assays of ALK-positive NB cell lines treated with AZD1208 or PIMi. Data represent means ± s.d. of technical triplicates. **d**, **e** Analysis of *PIM1* levels by RT-qPCR in **d** CHLA-20 or **e** KELLY cells with shRNA-mediated knockdown of *PIM1*. NT, non-targeting. Data represent means ± s.d. of triplicate experiments. **f**, **g** 72-h dose–response assays following brigatinib or ceritinib exposure in **f** CHLA-20 cells or **g** KELLY cells treated with *PIM1*-targeting or NT shRNA. Data points represent means ± s.d. of duplicate **f** or triplicate **g** experiments. ED$_{50}$ values were compared by an unpaired Student's *t*-test. Source data for 3a and 3d-g are provided as a Source Data file.

response to PIM inhibitors is independent of ALK status (Supplementary Fig. 7c).

**PIM1 is a therapeutic target in combination with ALK.** In order to determine whether inhibition of PIM1 may enhance the efficacy of ALK inhibitors in ALK-positive NB, CHLA-20 (*MYCN*-WT) and KELLY (*MYCN*-amplified) cells were transduced with a *PIM1*-targeting shRNA, achieving an approximate 50% reduction in *PIM1* expression as confirmed by RT-qPCR (Fig. 3d, e). *PIM1* knockdown increased the sensitivity of cells to brigatinib and ceritinib, indicated by a significant decrease in $ED_{50}$ concentrations after 72-h of treatment ($p < 0.0005$, unpaired Student's *t*-test, Fig. 3g) (Fig. 3f, g).

Recent human dose-escalation studies have displayed general tolerability for the PIM inhibitor AZD1208[47], which prompted the assessment of combined ALK and PIM1 inhibition in this study. To this end, cell viability following 72-h exposure to AZD1208 in combination with brigatinib or ceritinib in SH-SY5Y, CHLA-20 and KELLY cells using dose–response matrices in log-scale format was analyzed. The drug interactions were characterized using the Bliss Independence model described elsewhere[48]. A wide range of combination index (CI) values were determined across the concentration ranges for all cell lines and both ALK inhibitors, but for the most part CI values were <1, indicative of mild synergy between ALK and PIM inhibition (Fig. 4a). These findings were extended to other ALK-positive NB cell lines by assessing cell viability after treatment with fixed concentrations of brigatinib and AZD1208 (Fig. 4b). CI values ranged from 0.59 to 0.95, indicating variable levels of synergy between cell lines. The cellular consequences of drug interaction were then determined by analysis of apoptosis in several cell lines treated with brigatinib or AZD1208, alone and in combination. In 3/4 cell lines tested, synergy was observed between the drugs (with an additive effect observed in the fourth cell line tested) in terms of the apoptotic cell fraction detected after 48-h treatment (Fig. 4c).

**Combined inhibition of ALK and PIM in ALK-positive PDX.** To assess the efficacy of combined ALK and PIM inhibition in vivo, the COG-N-426 × (ALK$^{F1245C}$; *MYCN*-WT) and COG-N-453 × (ALK$^{F1174L}$; *MYCN*-amplified) PDX models of high-risk NB were employed. COG-N-426x is derived from the same primary patient tumor as the COG-N-426 cell line, although COG-N-426x cells were directly grafted from human to mouse without intermediary in vitro culture. As ceritinib is currently undergoing phase I assessment in patients with ALK-positive pediatric malignancies including NB (NCT02780128; NCT01742286), this ALK TKI was selected for in vivo investigation.

NOD *scid* gamma mice were treated daily with either vehicle (0.5% hydropropyl methylcellulose; 0.5% Tween-80), single-agent ceritinib (30 mg/kg), single-agent AZD1208 (15 mg/kg) or both agents in combination (Supplementary Fig. 8a). In the COG-N-426x model, single-agent treatment with ceritinib or AZD1208 led to a delay in tumor growth relative to vehicle treatment, although not significantly different from one another (Fig. 5a). However, the combination of ceritinib and AZD1208 led to a significant reduction in tumor volume at day 14 relative to either agent alone ($p < 0.05$, Mann–Whitney test). Kaplan–Meier survival analysis showed a significant increase in event-free survival (EFS) for animals treated with the combination of ceritinib and AZD1208 (median EFS = 29.5 days) relative to single-agent ceritinib (median EFS = 19 days; $p < 0.01$, Log-rank test) or AZD1208 (median EFS = 15 days; $p < 0.01$, Log-rank test, Fig. 5b), where an event was defined as a tumor reaching 15 mm in any direction. All compounds were well-tolerated, with no

significant decrease in body weight or lethal toxicity observed (Fig. 5c).

Residual COG-N-426x tumors harvested at the experimental end-point recapitulated histological features of NB, with H&E staining showing small, round, monomorphic cells with nuclear hyperchromasia and scant cytoplasm (Supplementary Fig. 8b). Consistent with the superior efficacy of combination treatment, tumor immunostaining showed a significant reduction in cell proliferation as indicated by positivity for Ki67 (Supplementary Fig. 8c; $p < 0.001$, unpaired Student's *t*-test) in combination-treated tumors versus single-agent ceritinib-treated tumors.

The superior efficacy of the combination therapy was confirmed in a second cohort of mice bearing *MYCN* + COG-N-453x NB xenografts (Fig. 5d, e). Tumor volume was significantly reduced at day 8 in combination-treated mice relative to mice treated with single-agent ceritinib or AZD1208 ($p < 0.05$, Log-rank test) (Fig. 5d). Kaplan–Meier survival analysis showed a significant increase in event-free survival (EFS) for COG-N-453x mice treated with the combination (median EFS = 31.5 days) relative to single-agent ceritinib (median EFS = 9 days; $p < 0.05$, Log-rank test) or AZD1208 (median EFS = 18 days; $p < 0.05$, Log-rank test) (Fig. 5e). These observations confirm the efficacy and tolerability of combined ALK and PIM1 inhibition in xenograft models of high-risk, ALK-positive NB independent of *MYCN* status.

**PIM1 is upregulated by ALK inhibition in PDX.** Analysis of *PIM1* mRNA expression in residual COG-N-426x tumors at the experimental end-point by RT-qPCR showed a significant increase in *PIM1* expression in ceritinib-treated animals relative to animals treated with AZD1208 or vehicle (DMSO) ($p < 0.05$, unpaired Student's *t*-test), while the combination of AZD1208 and ceritinib showed levels of *PIM1* comparable to those in vehicle-treated animals (Fig. 5f). This is in agreement with a gene expression dataset previously reported by Lambertz et al.[49] showing that in ALK-positive NB cell lines treated with 0.32 μM of the preclinical ALK inhibitor NVP-TAE684 for 6 h, *PIM1* is significantly upregulated relative to vehicle-treated cells in 4/5 cell lines tested (Supplementary Fig. 8d). In addition, the magnitude of gene upregulation appears to be independent of endogenous *PIM1* expression levels (Supplementary Fig. 8e).

## Discussion
The majority of patients undergoing treatment with molecularly targeted anti-cancer compounds acquire resistance after an initial response. For example, therapeutic resistance to ALK inhibitors is well documented for ALK-positive NSCLC and is known to arise through both ALK-dependent and -independent mechanisms[20]. ALK inhibitors are undergoing clinical assessment for ALK-positive pediatric malignancies including NB[50,51], and growing evidence from preclinical studies indicates that both de novo and acquired resistance will be of concern[13,16–19]. In this study, we proactively investigated mechanisms of resistance to ALK inhibitors using CRISPRa screens in NB cell lines harboring recurrent point mutations in the ALK tyrosine kinase domain, including those previously shown to confer endogenous resistance to the ALK inhibitor crizotinib.

Importantly, a genome-wide CRISPRa screening approach allowed us to fully explore all potential resistance mechanisms in an unbiased manner. Our work has expanded on the findings of previously published studies investigating ALK inhibitor resistance mechanisms in NB[13,52]; AXL activation was identified by a phospho-proteomic assay in NB cell lines rendered resistant to ALK inhibitors through continuous exposure to increasing concentrations of drugs[52], whereas *MYCN* overexpression was

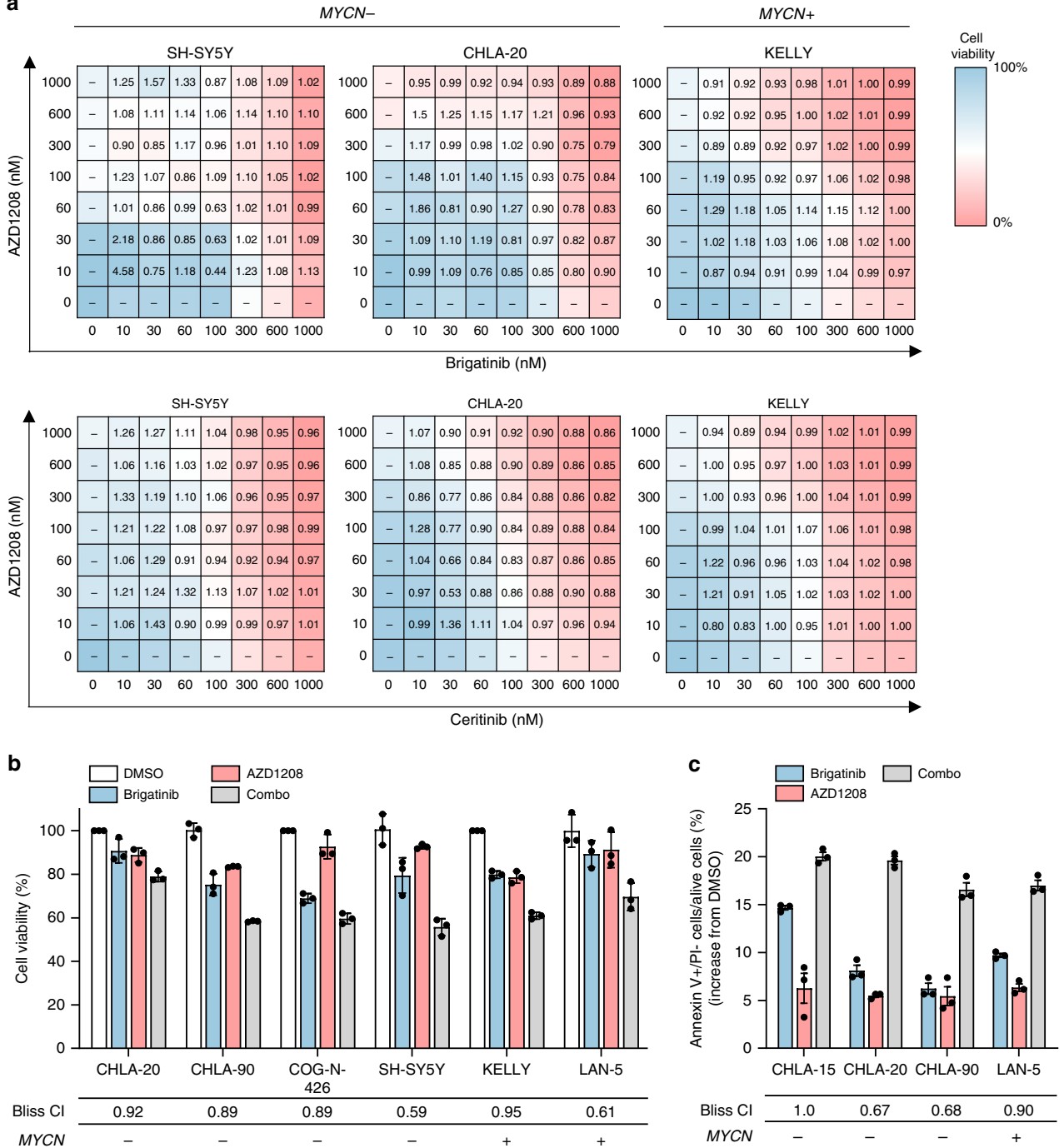

**Fig. 4** ALK and PIM inhibitors exhibit mild synergism in ALK-positive NB cell lines. **a** Dose–response matrices of brigatinib vs. AZD1208 and ceritinib vs. AZD1208 in SH-SY5Y, CHLA-20, and KELLY cell lines. Heat maps visually represent inhibition of cell viability after 72 h of treatment, and numbers indicate Bliss combination index (CI) values for each dose pair. Data points are representative of two independent experiments. **b** Assessment of the interaction between brigatinib and AZD1208 in a panel of ALK-positive NB cell lines using the Bliss Independence model. CI < 1 indicates synergism, CI = 1 indicates additivity and CI > 1 indicates antagonism. Data points represent means ± s.d. of triplicate experiments. Brigatinib was used at the approximate $ED_{10}$ concentration for each cell line $ED_{10}$ (CHLA-20) = 30 nM; $ED_{10}$ (CHLA-90) = 30 nM, $ED_{10}$ (COG-N-426) = 30 nM, $ED_{10}$ (SH-SY5Y) = 30 nM, $ED_{10}$ (KELLY) = 30 nM, $ED_{10}$ (LAN-5) = 3 nM). **c** Apoptosis analysis of ALK-positive NB cell lines treated with DMSO, brigatinib (1 μM), AZD1208 (1 μM), or a combination of the two (combo) for 48 h. Drug interaction was assessed using the Bliss Independence model, as described above. Data points represent means ± s.d. of triplicate experiments. Source data for this figure are provided as a Source Data file.

reported as a resistance mechanism in another study[13]. Of note, whilst *MYC* was identified as a resistance driver in our screen, neither *MYCN* nor *AXL* were among the resistance driver candidates identified. This may be due to insufficient overexpression of *MYCN* or AXL induced by the CRISPR-SAM system as

variable overexpression levels were observed for different genes in our study. Ultimately, the in vitro studies conducted to date are potentially predictive of resistance mechanisms in patients, but until more children with NB have been treated with ALK inhibitors and biopsy material is taken for study at relapse, CRISPR

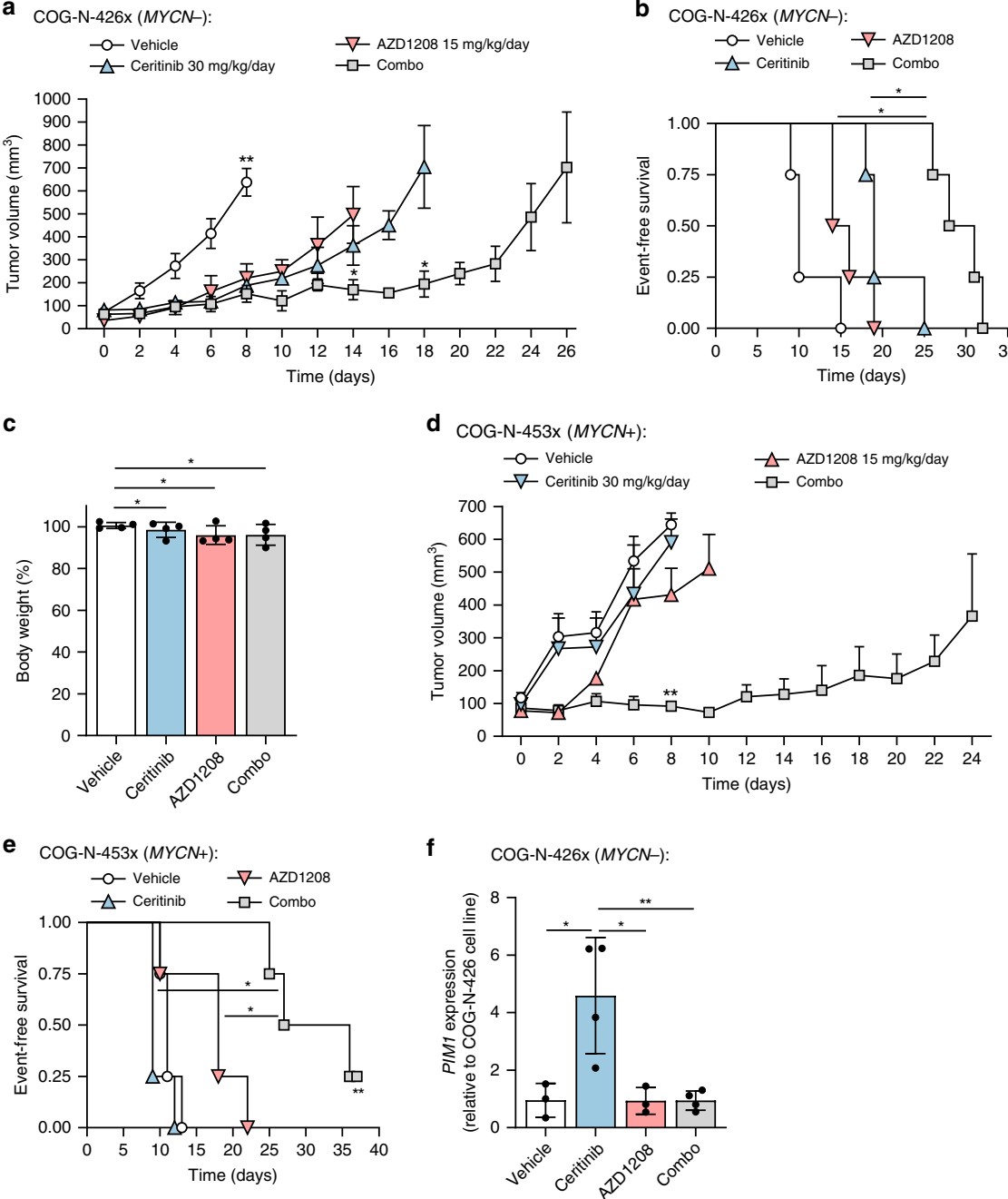

**Fig. 5** Efficacy of ceritinib and AZD1208 in driven patient-derived NB xenografts. **a** Tumor volume over time in COG-N-426x PDX mice administered daily with vehicle (0.5% hydroxypropyl methylcellulose, 0.5% Tween-80), ceritinib (30 mg/kg), AZD1208 (15 mg/kg), or ceritinib and AZD1208 (combo). Data points represent means ($n = 4$) ± s.e.m. and are shown until death of the first animal within each treatment group. *$p < 0.05$; **$p < 0.005$ (Mann–Whitney test). **b** Kaplan–Meier event-free survival according to each treatment group for tumor-bearing COG-N-426x mice, where survival is defined as the time taken for tumors to reach 15 mm diameter. *$p < 0.01$ (Log-rank test). **c** COG-N-426x mouse body weight at the experiment end-point relative to baseline per treatment group. Data points represent means ($n = 4$) ± s.d. *$p > 0.05$ (unpaired Student's $t$-test). **d** Tumor volume over time for COG-N-453x PDX mice administered daily with vehicle (0.5% hydroxypropyl methylcellulose, 0.5% Tween-80), ceritinib (30 mg/kg), AZD1208 (15 mg/kg), or ceritinib and AZD1208 (combo). Data points represent means ($n = 4$) + s.e.m. and are shown until death of the first animal within each treatment group. **$p < 0.005$ (Mann–Whitney test). **e** Kaplan–Meier survival curve per treatment group for tumor-bearing COG-N-453x mice, where survival is defined as the time taken for tumors to reach 15 mm diameter. The study was terminated on day 37, thus censoring one mouse (**). *$p < 0.01$ (Log-rank test). **f** PIM1 mRNA levels at the experimental end-point for COG-N-426x tumor-bearing mice per treatment group. Data points represent means ($n = 4$) ± s.d. of triplicate experiments, *$p < 0.05$; **$p < 0.01$ (unpaired Student's $t$-test). Source data for this figure are provided as a Source Data file.

screens, while having their caveats, offer the best approach for global, unbiased identification of resistance mechanisms. CRISPRa screens were conducted, employing the second-generation ALK inhibitors brigatinib and ceritinib at both $ED_{50}$ and $ED_{75}$

concentrations to enable discrimination between potential compound- and concentration-dependent resistance mechanisms. As expected, we observed the greatest number of resistance genes at the higher ($ED_{75}$) concentrations of brigatinib and ceritinib in

SH-SY5Y and CHLA-20 cells. Distinct sets of resistance genes were observed amongst different concentrations of ALK inhibitors, with only three genes (*MET*, *PIM1*, and *SAGE1*) common to both inhibitors at both $ED_{50}$ and $ED_{75}$ concentrations. Activation of c-MET has previously been shown to confer resistance to alectinib in ALK-positive NSCLC[31], consistent with a partial functional redundancy of c-MET and ALK due to convergence on several pro-survival signaling pathways[20]. PIM1 is a serine/threonine kinase, the expression of which is controlled by the JAK/STAT pathway, and has also been associated with resistance to chemotherapy[33–35] and molecularly targeted agents[36,37] in several cancer types, but to our knowledge not previously for NB. Of the 16 resistance genes identified in brigatinib and ceritinib-treated NB cells at the $ED_{75}$, two are known to be activated downstream of ALK, namely *KRAS* and *PIK3CD* (PI3Kδ). Copy number gain and mutational activation of *KRAS* at codon 12 has been shown to confer resistance to crizotinib and ceritinib in ALK-positive NSCLC[24,53]. Similarly, mutational activation of *PIK3CA* is reported as a resistance mechanism to alectinib and ceritinib in these patients[54,55]. Interestingly, we identified a resistance gene (*MFSD2A*)[56] encoding a sodium-dependent transporter of fatty acids expressed in brain endothelium, in both SH-SY5Y and CHLA-20 cells treated with brigatinib or ceritinib at $ED_{75}$ concentrations, most likely functioning as an efflux pump for ALK inhibitors although this remains to be investigated further. Encouragingly, gene set enrichment analysis of the putative resistance genes showed an enrichment for genes involved in negative regulation of cell death.

*MET* was the only putative resistant gene common to all ALK inhibitors. Given that crizotinib is a potent inhibitor of c-MET[31] and can overcome ALK inhibitor resistance driven by activation of MET in NSCLC[32], we chose to focus instead on *PIM1* whereby high *PIM1* gene expression levels were found to be associated with advanced, high-risk disease and poor survival outcomes on analysis of published datasets[28,29]. In addition to validating *PIM1* as a resistance gene in NB cell lines, we sought to determine whether *PIM1* induces resistance to ALK inhibition in another ALK-driven pediatric cancer, namely ALCL, since full-length ALK and NPM-ALK have been shown to activate common downstream pathways such as Ras/MAPK, PI3K/AKT and JAK/STAT[50]. Indeed, overexpression of *PIM1* in ALK-positive ALCL cell lines decreased sensitivity to brigatinib and ceritinib, consistent with results published previously demonstrating robust synergy between a small-molecule pan-PIM inhibitor and crizotinib in ALCL cell lines[40]. Therefore, further studies investigating the potential for combined PIM and ALK inhibition in other ALK-positive malignancies are warranted.

We assessed the in vitro responses of both ALK-positive and ALK-negative NB cells to several small-molecule pan-PIM kinase inhibitors and found that cells were relatively insensitive after 72 h of exposure. However, knockdown of *PIM1* by RNA interference sensitized cells to ALK inhibition and the combination of ALK inhibitors with AZD1208 demonstrated mild synergy, prompting us to investigate the combination of ceritinib and AZD1208 in vivo. To examine the clinical relevance of this drug combination, we employed two patient-derived models of high-risk NB harboring $ALK^{F1245C}$ or $ALK^{F1174L}$ mutations respectively. PDX models are perhaps more representative of the clinical scenario than are cell lines due to their relative genetic likeness to tumors in patients, as opposed to long-term cultured cell lines that have evolved further from the original malignancy. We observed a significant delay in tumor growth with the combination treatment relative to single-agent treatments in both models. Of note, PIM inhibition sensitized both *MYCN*-amplified and wild-type, *ALK*-driven neuroblastoma cells to ALK inhibitors both in vitro and in vivo, suggesting co-inhibition of PIM and

ALK as a viable strategy to enhance the efficacy of ALK inhibitors. It has previously been shown that ALK and MYCN are part of a positive feedback loop whereby ALK regulates expression of MYCN through repression of HPB1[57]. However, our data, particularly the in vivo PDX data suggest that combined PIM and ALK inhibition is effective independent of MYCN status. Moreover, *PIM1* mRNA levels were significantly elevated in COG-N-426x tumors treated with ceritinib relative to vehicle at the experimental end-point, thus providing in vivo evidence of *PIM1* as a resistance gene. In summary, our data confirm that PIM1 induces resistance to ALK inhibitors in NB cell lines and also demonstrate the potential for combined pharmacological inhibition of ALK and PIM1 in patients with ALK-positive, high-risk NB.

## Methods

**Cell lines and cell culture.** The neuroblastoma cell lines CHLA-15, CHLA-20, CHLA-42, CHLA-90, CHLA-95, CHLA-171, COG-N-426 (Felix), LA-N-5, LA-N-6, NB-1643, NB-EBC1, SK-N-BE(1), SK-N-BE(2), SK-N-FI, SMS-KAN, SMS-KCR, and SMS-LHN were obtained from COG. CHP-134, IMR-32, KELLY, LA-N-1 and SH-SY5Y were obtained from the European Collection of Authenticated Cell Cultures (ECACC). GI-ME-N, NBL-S and NGP were obtained from German Collection of Microorganisms and Cell Cultures (DSMZ) and 293FT was obtained from Thermo Fisher Scientific. CHLA-15, CHLA-20, CHLA-42, CHLA-90, CHLA-95, CHLA-171, COG-N-426 (Felix), NB-1643, NB-EBC1 and NBL-S cells were cultured in IMDM (Gibco, Cat#21980032) supplemented with 20% FBS, 1% insulin-transferrin-selenium (ITS; Gibco, Cat#41400045) and 1% penicillin/streptomycin (PS). CHP-134, GI-ME-N, IMR-32, KELLY, LA-N-1, LA-N-5, LA-N-6, NGP, SK-N-BE(1), SK-N-BE(2), SK-N-FI, SMS-KAN, SMS-KCNR, and SMS-LHN cells were cultured in RPMI 1640 medium (Gibco, Cat#21875091) supplemented with 10% FBS, 1% insulin-transferrin-selenium (ITS) and 1% penicillin/streptomycin (PS). SH-SY5Y and 293FT cells were cultured in DMEM (Gibco, Cat#41966029) supplemented with 10% FBS and 1% PS. Cells were grown at 37 °C in a humidified incubator with 5% $CO_2$. All cells were mycoplasma-free and subjected to quarterly in-house testing using the EZ-PCR Mycoplasma Detection Kit (Geneflow, Cat#K1-0210).

**Genotyping of cell lines for *ALK* status.** Total RNA was extracted using the RNeasy Plus Mini Kit (Qiagen, Cat#74134). RNA was reverse transcribed using iScript Reverse Transcription Supermix (Bio-Rad, Cat#1708840). The kinase domain of ALK was PCR-amplified from 10 ng cDNA using the Q5 High-Fidelity PCR Kit (NEB) with primers F: CGGCATCATGATTGTGTACC; R: GTTGCTTTTGCTGGGGTATG. Amplicons were resolved by agarose gel electrophoresis and extracted using the QIAquick Gel Extraction Kit (Qiagen, Cat#28706). DNA was Sanger sequenced with a custom sequencing primer, F: ACCTCGACCATCATG. Chromatograms were analysed in SnapGene Viewer using that of CHP-134 as reference, since its ALK coding sequence is identical to RefSeq NM_004304 (NCBI-BLAST).

**Cell viability assays.** Cells were seeded in black/clear bottom 96-well plates (Greiner Bio-One, Cat#655209) at densities that resulted in near-confluency after 96 h. After 24 h, the media was aspirated and 200 μL fresh media was added containing compounds in log-scale concentrations (0, 1, 3, 10, 30, 100, 300, 1000, 3000 nM) in technical triplicate. DMSO concentration was fixed at 0.3% for all cells. Plates were sealed with AeraSeal breathable film (Excel Scientific, Cat#BS-25) to reduce evaporation and cells were incubated at 37 °C. After 68 h, 120 μL CellTiter-Blue cell viability reagent (Promega, Cat#G8082) diluted 1:5 with media was added to each well and cells were incubated at 37 °C for 4 h. After 4 h, fluorescence was read on a SpectraMax i3 microplate reader (Molecular Devices). PBS was used to calculate background signal for subtraction from all measurements. Normalized dose–response curves were generated by non-linear regression and $ED_{50}$ values calculated with GraphPad Prism 7 software (GraphPad Software Inc).

**Colony formation assays.** Cells were seeded into six-well plates at $10^3$ cells per well. After 24 h, compounds were added at the following log-scale concentrations: 0, 30, 100, 300, 1000, 3000 nM (0.3% DMSO) and media was replaced every three days. Cells were cultured for 12 days, after which each colony in the DMSO control comprised at least 50 cells. Colonies were fixed with 10% neutral buffered formalin, stained with 0.05% (w/v) crystal violet (25% methanol), washed and imaged on an ImageQuant LAS-4000 imaging system (GE Life Sciences).

**Synergy experiments.** For dose–response curves, cells were treated with log-scale concentrations of ALK inhibitors as described above in addition to fixed concentrations of AZD1208 as indicated. Potential synergy between ALK inhibitors

and AZD1208 was evaluated by calculating the combination index (CI) based on the Bliss Independence model[48]. The CI can be calculated with the following equation: $CI = (Ea + Eb - ((Ea*Eb))/Eab$, where Ea indicates the viability effect of drug A (ALK inhibitor) and Eb indicates the viability effect of drug B (AZD1208) and Eab indicates the viability effect of the drug combination. CI < 1 indicates synergism, CI = 1 indicates additivity and CI > 1 indicates antagonism. For dose–response matrices, cells were treated with log-scale concentrations of each compound in $8 \times 8$ grids and DMSO concentration was maintained at 0.3%. Cell viability was measured with CellTiter-Blue as described above and data were normalized to the average of the untreated well. All synergy experiments were performed in technical triplicate.

**Cloning of guide sequences**. Promoter-targeting guide sequences for the 15 genes used to confirm CRISPR SAM activity were designed with the Cas9 Activator Tool (sam.genome-engineering.org/database) and the oligonucleotides are shown in Supplementary Table 5. Guide sequences for targeting of the screen candidate genes were obtained from the Addgene depositor datasheet (www.addgene.org/pooled-library/zhang-human-sam-v1) and modified by the addition of BsmBI recognition sites (Supplementary Table 6). Oligonucleotide sequence for generation of 20 bp non-targeting (scrambled) guide sequence were kindly shared by Dr Feng Zhang. Double-stranded sequences were generated by phosphorylating and annealing oligonucleotides with T4 polynucleotide kinase (NEB, Cat#M0201) and T4 DNA ligase (NEB, Cat#M0202), respectively. Guide sequences were cloned into the lenti sgRNA(MS2)_zeo backbone (Addgene, Cat#61427) by Golden Gate assembly with BsmBI (NEB, Cat#R0580) and amplified in NEB Stable competent *E. coli*. A full protocol is available at www.sam.genome-engineering.org/protocols.

**Production of lentivirus and cell transduction**. Low-passage 293FT cells were cultured in DMEM with 5% FBS. Cells were seeded at 40% confluency in six-well plates for small-scale production or 15 cm dishes for larger scale production. Once confluency reached 80–90%, cells were transfected with pMD2.G (Addgene, Cat#12259), psPAX2 (Addgene, Cat#12260) and transfer plasmid at equimolar ratios using TransIT-293 transfection reagent (Mirus Bio, Cat#MIR2704) according to the manufacturer's instructions. The media containing lentivirus was collected after 60 h. Cellular debris was removed by centrifugation at $500 \times g$ and the media was passed through a 0.45 μm polyethersulfone (PES) filter. The supernatant was stored at −80 °C. To generate stable cell lines, cells were seeded into culture vessels in media supplemented with 5% FBS, grown to near-confluency and then transduced with lentiviral media. After 20 h, cells were split into media containing antibiotics as follows: 10 μg/mL blasticidin for 7 days, 300 μg/mL hygromycin for 7 days, 500 μg/mL zeocin for 5 days or 1.5 μg/mL puromycin for 5 days. Antibiotic concentrations were determined by kill curve analysis.

**Generation of CRISPR SAM cell lines**. CHLA-20 and SH-SY5Y cells were co-transduced with lentiviruses carrying the dCas9-VP64_blast (Addgene, Cat#61425) and lenti MS2-p65-HSF1_hygro (Addgene, Cat#61426) constructs at MOI~0.2 and selected as described above. To confirm CRISPR SAM activity, cells were transduced with lentivirus carrying the lenti sgRNA(MS2)_zeo construct (Addgene, Cat#61427) into which guide sequences were cloned and overexpression of the target genes was confirmed by RT-qPCR.

**RT-qPCR analysis**. Total RNA was isolated from cell lines using the RNeasy Plus Mini Kit (Qiagen, Cat#74134). In total, 1 μg RNA was reverse transcribed using iScript Reverse Transcription Supermix (Bio-Rad, Cat#1708840) and 20 ng cDNA template was used for qPCR. cDNA was amplified with PowerUp SYBR Green Master Mix (Thermo Fisher Scientific, Cat#A25918) and 600 nM primers on a QuantStudio 6 Flex Real-Time PCR System (Thermo Fisher Scientific) under standard cycling conditions. Relative quantification ($\Delta\Delta C_T$) analysis was conducted with normalization to *GAPDH*. All reactions were performed in technical triplicates and each plate included no-RT and no-template controls. Primer sequences for all RT-qPCR experiments are listed in Supplementary Table 7.

**CRISPR SAM screens**. The zeocin-resistant pooled gRNA library v1 (Addgene, Cat#1000000057) was packaged into lentivirus in 15 cm plates, as described above. CHLA-200 and SH-SY5Y cells stably expressing dCas9-VP64 and MS2-p65-HSF1 were seeded into 15 cm plates and transduced with the lentiviral gRNA library at MOI 0.3–0.4, with a minimum representation of 500 transduced cells per guide. After 20 h, cells were split into 500 μg/mL zeocin for 5 days and passaged every second day, maintaining >500 cells per guide. After 5 days of selection, $3.6 \times 10^7$ cells were harvested as a day 0 sample. The remaining cells were split into duplicate populations and cultured in the presence of vehicle (0.3% DMSO), brigatinib or ceritinib at $ED_{50}$- and $ED_{75}$-equivalent concentrations as determined by 72-h dose–response assay. Cells were maintained below 80% confluency and harvested after 14 days of treatment for genomic DNA extraction.

**Preparation of HiSeq libraries**. Genomic DNA was extracted with the QIAamp DNA Blood Maxi Kit (Qiagen, Cat#51194). The gRNA regions were PCR-amplified for 22 cycles with Herculase II Fusion DNA Polymerase (Agilent, Cat#600677) in

23 replicate reactions, with a total input of 230 μg DNA corresponding to 500X library coverage. Each 100 μL reaction comprised Herculase II reaction buffer (2 mM $Mg^{2+}$), 1 mM dNTPs, 2% DMSO, 250 nM pooled forward primer, 250 nM specific reverse primer and 10 μg template DNA. The Illumina-compatible primers contained P5/P7 adapters, a staggered region (forward primers only) and 8 bp index barcodes (Supplementary Table 8). PCR products were pooled and 400 μL was ethanol precipitated, resuspended in 100 μL water and resolved on a 3% agarose gel at 100 V for 5 h. The 270–280 bp amplicon was isolated with the QIAquick Gel Extraction Kit (Qiagen, Cat#28706) according to the manufacturer's protocol, except the QC buffer was incubated at room temperature. Products were tested for concentration and specificity with High sensitivity D1000 ScreenTape and qPCR using KAPA Library Quantification Kit, multiplexed, spiked with PhiX Control v3 library (Illumina, Cat#FC-110-3001) and run on an Illumina HiSeq 2500 on Rapid Run $1 \times 100$ bp mode for 115 cycles.

**NGS gene enrichment analysis**. Raw FASTQ files were downloaded from the Bauer Core server using FileZilla. Initial QC was conducted with FastQC (Babraham Bioinformatics, Cambridge, UK) to assess the general quality of the sequence runs. Raw FASTQ sequencing files were demultiplexed with bcl2fastq2 v2.2, then matched to the guide sequences from the library files using the MAGeCK count function. Due to the absence of non-targeting guides in this library, an analysis to control for the empirical false positive rate could not be performed. Read counts per gRNA were calculated by averaging gRNA read counts of two biological replicates per screen condition and normalizing to total gRNA read count. Normalized read counts in ALK inhibitor-treated cell populations were log-transformed and compared with those in DMSO-treated cell populations to identify gRNAs that were preferentially enriched under ALK inhibitor conditions. An arbitrary threshold of >1.5-fold enrichment was applied and genes with multiple gRNAs exceeding this threshold were considered to be enriched. To improve stringency, only genes enriched in both brigatinib and ceritinib screens at a given concentration ($ED_{50}$ or $ED_{75}$) were considered as candidates.

**Gene set enrichment analysis**. Hallmark and gene ontology gene expression datasets were downloaded from the molecular Signatures database (MSigDB) v6.2[27] and analyzed with GSEA v3.0 (www.broadinstitute.org/gsea)[58,59]. Enrichment was carried out by calculating overlaps between MSigDB datasets and putative resistance genes in SH-SY5Y cells treated with brigatinib and ceritinib at $ED_{50}$ and/or $ED_{75}$ concentrations.

**Analysis of public datasets**. All candidate genes identified from the CRISPR SAM screens were investigated by Kaplan–Meier event-free survival analysis with microarray data from primary neuroblastoma patient cohorts using R2: Genomics Analysis and Visualization Platform (http://r2.amc.nl). The following cohorts were analysed: Kocak ($n = 476$) [accession: GSE45547][29] and SEQC ($n = 498$) [accession: GSE49710][28]. The cut-off method was selected as scan to determine the optimal threshold for each gene, and significance was assessed by log-rank test. The *p*-values were corrected for multiple testing using the Bonferroni method. The hazard ratio was determined through a Cox proportional hazards model using the coxph function in the survival package in R. The data analyzed in Supplementary Figs. 8d and e can be accessed through ArrayExpress (accession number E-MTAB-3205)[49].

**Immunoblot analysis**. Adherent cells were washed and lysates prepared with pre-chilled RIPA Lysis and Extraction Buffer (Thermo Fisher Scientific, Cat#89900) supplemented with 1% Halt Protease and Phosphatase Inhibitor Cocktail (Thermo Fisher Scientific, Cat#78440). Around 50 μg protein lysate per sample was resolved by SDS-PAGE and transferred to a 0.45 μm PVDF membrane for 2.5 h at 80 V according to the manufacturer's protocol (Bio-Rad). The membrane was blocked in 5% BSA and blotted with antibodies overnight at 4 °C. Primary antibodies were as follows and diluted 1:1000 unless specified otherwise: anti-BAD (CST, Cat#9292), anti-phospho-BAD (Ser112) (CST, Cat#9291), anti-BAX (CST, Cat#5023), anti-BCL-2 (CST, Cat#4223), anti-PIM1 (CST, Cat#2907), anti-PRAS40 (CST, Cat#2691), anti-phospho-PRAS40 (Thr246) (CST, Cat#13175), anti-AKT (CST, Cat#9272), anti-pAKT (CST, Cat#9271), anti-ALK (CST, Cat#3633), anti-pALK (CST, Cat#3341), anti-GAPDH (CST, Cat#97166; diluted 1:10,000), and anti-α-tubulin (CST, Cat#T9026; diluted 1:10,000). Membranes were then incubated with secondary antibodies for 1 h at room temperature. Secondary antibodies were as follows and diluted 1:10,000: anti-mouse HRP-immunoglobulins (Dako, Cat#P0260) and anti-rabbit HRP-immunoglobulins (Dako, Cat#P0448). If required, membranes were stripped once with stripping buffer (2% SDS, 62.5 mM Tris HCl [pH 6.8], 0.8% β-mercaptoethanol) and re-blocked. Membranes were developed with Immobilon Western Chemiluminescent HRP Substrate (Merck Millipore) and imaged with an ImageQuant LAS-4000 imaging system (GE Life Sciences). Raw, unprocessed blots are presented for Fig. 2c in the Source Data file.

**Mice and animal housing**. NOD *scid* gamma (NSG) mice were obtained from Charles River and housed in groups of 2–5. Animal work was carried out under UK Home Office licence P4DBEFF63 according to the Animals (Scientific Procedures) Act 1986, and was approved by the University of Cambridge Animal Welfare and

Ethical Review Body (AWERB). We have complied with all relevant ethical regulations for animal testing and research in the UK.

**Xenograft study**. COG-N-426x and COG-N-453x patient-derived xenograft (PDX) cells were obtained from the Childhood Cancer Repository maintained by the Children's Oncology Group (COG). Cells were suspended in Matrigel diluted 1:2 with PBS and $3 \times 10^5$ cells (300 µL) injected into the left flank of NSG mice at 6–8 weeks of age. Tumors were measured daily with manual calipers and tumor volumes estimated using the modified ellipsoid formula: $V = ab^2/2$, where a and b (a > b) are length and width measurements. Once tumors reached approximately 75 mm³, mice were randomly allocated into four treatment groups ($n = 4$ per group) and treated daily with the following agents by oral gavage at 10 µL per gram body weight: vehicle (0.5% hydroxypropyl methylcellulose; 0.5% Tween-80), ceritinib (30 mg/kg), AZD1208 (15 mg/kg) or the combination of ceritinib (30 mg/kg) and AZD1208 (15 mg/kg). Mice were euthanized once tumors reached 15 mm in any direction (defined as an event for event-free survival analysis) and tumors were fixed in 10% neutral-buffered formalin for 24 h. Tumors were paraffin-embedded and 10 µm sections were cut from central regions. COG-N-426x tissue sections were stained with hematoxylin and eosin or with antibodies against ALK (CST, Cat#3633) or Ki-67 (CST, Cat#12202). All antibodies were diluted 1:200 and incubated on slides overnight at 4 °C. Slides were then incubated with secondary biotinylated antibody (Vector Laboratories, Cat#BA-1000) in 2% goat serum for 30 min at room temperature. Slides were washed and incubated with VECTASTAIN Elite ABC reagent (Vector Laboratories, Cat#PK-6101) for 45 min followed by incubation with Vector DAB HRP substrate (Vector Laboratories, Cat#SK-4100) until the desired stain intensity was achieved. Slides were counterstained with Mayer's hemalum solution (Carl Roth, Cat#T865.1) and mounted with Aquatex reagent (Sigma-Aldrich, Cat#1085620050). RNA was extracted from 10 µm sections of COG-N-426x tumors using the PureLink FFPE RNA Isolation Kit and treated with RQ1 RNase-Free DNase before reverse transcription and qPCR as described earlier. In total 20 ng cDNA was used per qPCR reaction.

**Analysis of apoptosis**. Cells were collected by centrifugation, washed once in PBS, then resuspended in 100 µL Annexin V Binding Buffer containing 5 µL/reaction APC-Annexin V (BioLegend, Cat#640920) and incubated for 30 min at room temperature. APC-Annexin V was then removed from the cells following centrifugation, washed once in Annexin V Binding Buffer (Thermo Fisher Scientific, Cat#V13246), and cells resuspended in 100 µL Annexin V Binding Buffer containing 1 mg/mL propidium iodide (Sigma-Aldrich, Cat#P4170-10MG). Data were acquired using a BD Accuri C6 flow cytometer and analyzed using FlowJo V10 software. Cells were gated according to physical parameters in order to discard cell debris (FCS/SSC) and cell clumps (width/area).

**Statistical analysis**. Methods for statistical analysis are specified in the relevant Fig. captions. All Student's $t$-tests, ANOVA models, correlation analyses and Kaplan–Meier survival analyses in mice were conducted with GraphPad Prism 7 software. Heat maps for cell viability were produced in Excel 2016. Gene set enrichment analysis was conducted in GSEA v3.0 software as described above. Kaplan–Meier survival analyses in human patients from published gene expression datasets were generated in R2: Genomics Platform as described above.

**Reporting summary**. Further information on research design is available in the Nature Research Reporting Summary linked to this article.

## Data availability

All relevant data supporting the key findings of the study are available within the article and its Supplementary Information files or from the corresponding author upon reasonable request. The source data underlying Figs. 2, 3a, 3c–g, 4 and 5, and Supplementary Figs. 1a, d, 3, 4, 5a–d, 5f–g, 6, 7 and 8c-e are provided as a Source Data file. A reporting summary for this Article is available as a Supplementary Information file. Publicly available microarray data used for Kaplan-Meier event-free survival analysis is available from the NCBI gene expression omnibus with accession codes: GSE45547[29] and GSE49710[28]. The data analyzed in Supplementary Figs 8d and e can be accessed through ArrayExpress with accession code E-MTAB-3205[49].

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

## Acknowledgements

This research is supported with funding from Children with Cancer UK (CWCUK; 16-209) awarded to S.D.T. and L.C.L., R.M.T. was supported by the CWCUK grant and L.C.L. was in receipt of a Cancer Research UK Cambridge Centre Paediatric Programme PhD studentship. N.P., L.J., O.M., I.G.F.A., L.K. and S.D.T. are supported by a European Union Horizon 2020 Marie Skłodowska–Curie Innovative Training Networks (ITN-ETN) Grant, Grant No.: 675712. N.A.P. was supported by ERASMUS+. We are grateful to AstraZeneca for providing us with AZD1208 and Inflection Biosciences for providing us with PIMi. We thank Isaia Barbieri for critical suggestions on the manuscript, BBSCR NIHR Cell phenotyping hub for flow cytometry expertise, the Bauer Core Facility at Harvard University for sequencing, and Liew Jun Mun and Stephen P. Ducray for technical assistance. Funding for the COG/ALSF Childhood Cancer Repository was provided by Alex's Lemonade Stand Foundation.

## Author contributions

S.D.T., J.L., R.M.T., L.C.L. and N.P. designed the research; R.M.T., L.C.L., N.P., L.J., N.A.P., M.H., H.K.L., S.T., S.H., I.G.F.A., E.M. and P.K. performed the experiments; R.M.T., L.C.L., N.P., J.D.M., O.M., L.K. and S.D.T. analyzed data; R.M.T., L.C.L., N.P. and S.D.T. wrote the paper; C.P.R., S.M., J.P. and C.B.A. provided essential reagents, namely patient-derived xenografts (CPR) or PIM1 inhibitors (S.M., J.P., C.B.A.). R.M.T., L.C.L., N.P., L.J., C.P.R., G.A.A.B., N.A.P. M.H., J.D.M., H.K.L., E.M., S.M., J.P., C.B.A., O.M., I.G., P.K., S.T., S.H., J.L., L.J. and S.D.T. contributed intellectually to the editing of the manuscript and interpretation of data.

## Competing interests

The authors declare no competing interests.
