## [Peer Review File · Nature Communications]

Reviewers' Comments:

Reviewer #1:

Remarks to the Author:

The manuscript highlights PIM1 kinase as a resistance gene in ALK-positive neuroblastoma and suggests increased antitumor efficacy of combined ALK and PIM1 inhibition for high-risk neuroblastoma. The manuscript is well written, and hypothesis driven. PIM1 has been linked to chemotherapeutic resistance in other cancers (especially prostate and breast cancer), and it is not surprising to know that kinase has similar roles in neuroblastoma. The specificity for PIM providing resistance to ALK-driven cancers is a novel finding, but the underlying mechanisms allowing for survival (or synergistic cell death with PIM inhibition) could be explored further. The therapeutic strategy of combining ALK and PIM inhibitors for high-risk neuroblastoma holds potential but current manuscript lacks detail on the mechanisms by which these pathways interact and the relative importance of this mechanism for resistance to therapy, considering the mild synergy observed (albeit, this was better in vivo – which could speak to the role of PIM other aspects of tumor biology such as angiogenesis). In my opinion, the manuscript could be reconsidered for publication in nature communications if authors address the comments below in their revision.

Major comments

- 1) In contrast to the text stating that PIM1 is associated with worse survival in NB, the data in Supplemental Figure 3 appear to show that patients with high PIM1 (red) actually have considerably better overall survival compared to patients with low PIM1 (blue)?
- 2) It would be interesting to know the status of the other PIM isoforms (PIM2 and PIM3) in relation to NB outcome as well as the main cell lines used in this study, since they are known to compensate for each other.
- 3) In figure 5, PIM1 mRNA levels are increased in the xenograft model, albeit only in the ceritinib-treated group. Protein levels should also be assessed - IHC for several other factors in these tumors is presented in Supp Fig 5A; why is there no IHC for PIM1?.
- 4) Can the authors speculate on what is causing this increase in PIM1 levels in response to ceritinib, and why are PIM1 levels not similarly high in the combo-treated mice? How PIM can be acting as a mechanism of resistance if it's not being upregulated and is inherently low in some of the cell lines tested?
- 5) Some of the cell lines used are more sensitive to PIM inhibition than others, and the patterns of synergy manifest differently (i.e., in the CHLA-20 cells you need much higher concentrations of ALK inhibitor to see synergy, whereas in the SH-SY5Y cells you see synergy at relatively low concentrations of both ALK and PIM inhibitor). Do different ALK mutations respond to PIM inhibition/use PIM signaling as a resistance mechanism?
- 6) Although it's briefly mentioned, some further discussion on why the combination treatment doesn't appear to increase cleaved caspase-3 would be useful, particularly as the focus earlier in the manuscript states that PIM induces resistance by helping cells evade apoptosis. The manuscript could benefit from further experiments here, like assessing other mechanisms of cell death that are associated with PIM biology.

Minor comments

- 1) The order of main and supplementary figures/table should be continuous.
- 2) Figure 3 should include western blots for phospho-Bad (and/or other PIM targets) following PIM

shRNA knockdown to confirm that the pathway is inactive.

Reviewer #2:

Remarks to the Author:

Trigg and colleagues have submitted a manuscript which examines the potential genes that are implicated in ALK inhibitor resistance in ALK-positive neuroblastoma. The authors used CRISPR-Cas9 overexpression screens and treated these neuroblastoma cells with ALK inhibitors to identify genes that were overexpressed downstream to ALK. Among the genes that were identified and druggable, the authors identified PIM1. NB cells with PIM1 overexpression were found to be more resistant to ALK inhibitors. While PIM1 inhibition with the small molecule inhibitor AZD1208 did not have a marked effect on neuroblastoma cells, PIM1 silenced (shRNA) neuroblastoma cells were more sensitive to ALK inhibitors than cells without PIM silencing. Finally, the authors found that in vivo, the combination of ALK inhibitor with PIM1 inhibitor markedly inhibited tumor growth and increased survival in PDX models of ALK muted neuroblastoma. Overall, the paper is well written and well presented. The studies appear to be well performed and presented. In order for further consideration for publication, the following points should be addressed.

1 . While both MYCN amplified and non-amplified PDX were utilized in the in vivo experiments, only MYCN non-amplified cell lines were used for the in vitro studies (SH-SY5Y and CHLA-20). Please justify why MYCN amplified cell lines were not included in this aspect of the study, knowing the clinical importance of MYCN amplification. To this end, please incorporate more discussion on the relationship between MYCN and ALK, especially in light of a recent publication by the Speleman group in Oncogene, Dec 2018.

2. In the in vivo experiments which compare tumor volume in mice amongst the different treatment groups, the authors have used an inappropriate parametric statistical method (two way ANOVA) which falsely overestimates the statistical significance of these very small treatment groups. For these animal studies, a non-parametric statistical technique should be employed, such as Kruskal Wallis or Mann-Whitney.

3. In the animal studies, the authors use Kaplan-Meier survival analysis based on event free survival. It is not entirely clear, but is the only defining event of a mouse being discontinued from the study is to have a tumor reach 15 mm in size? If this is correct, this should also be included in the Methods section.

Minor issues:

On page 6, near the bottom, the text reads "(Supplementary Figures SC and 3D)" . it should read "(Supplementary Figures 3C and 3D)".

On Page 7, about halfway down the page, the text reads "upon overexpression of PIM1 (Figure 2C)". This is not Figure 2C, but rather Supplementary Figure 4C.

It is not clear why several figure panels are not discussed or referenced in the paper - for example, Supplementary Figure 4B and Figure 3B are not mentioned or referenced but they are in the figures. Please reconcile.

Reviewer #3:

Remarks to the Author:

In the submitted manuscript, "PIM1 kinase is a synergistic therapeutic target with ALK inhibitors in ALK high-risk neuroblastoma" Trigg and coworkers proactively try to identify future expected resistance mechanisms in ALK positive neuroblastoma. They employ a genome wide CRISPR-Cas9

overexpression screen in two cell lines positive for ALK followed by treatment with clinical FDA approved ALK inhibitors. From the treatment with ALK inhibitors the authors find enrichment of 25 genes including a couple of the usual suspects, such as PI3K, KRAS, MYC (although not clear which MYC) MET, FOX and ETVs. Only 3 genes were enriched by both inhibitors (MET, PIM1 and SAGE1). The authors focus further analysis on PIM1 and show that PIM1 overexpression induces resistance to ALK inhibitors and suggests that combined inhibition of ALK and PIM1 is a good strategy for the treatment of ALK-positive high-risk neuroblastoma. However, the synergy with ALK is very mild as suggested by the authors.

Comments

1. The order of the figures and supplementary figures should be rearranged according to the mentioned in the main text. No page or lane numbering included.
2. Reference for sentence "Anaplastic lymphoma kinase (ALK) is the most commonly mutated gene in NB,cases overall" reference 10-14, should only be "de Brouwer and Bresler" as they have investigated more than 700 and 1500 patients, respectively. Chen, Mosse, Georg and Janoueix-Lerosey do not discuss the most commonly mutated gene in NB neither do they have secure information regarding numbers of ALK mutations in NB.
3. Initially, error bars are missing in every cell viability assay shown in this manuscript. To make it easier for the reader a table should be included, which show the IC50 values of dose response of treated cell lines employed.
4. From earlier in vitro kinase assays it is clear that brigatinib/ceritinib inhibit ALK wild type/mutation kinase activity with low IC50 values in the range of 0.6 nM to single-digit nM. In different cell line models brigatinib/ceritinib inhibit ALK wild type and mutation ALK-F1174L driven proliferations/survival with 14 nM and 55 nM, respectively (Zhang et al., Clin Cancer Res. 2016 Nov 15;22(22):5527-5538. doi: 10.1158/1078-0432.CCR-16-0569; Cecon, M., Mologni L., Giudici, g., Piazza, R., Pirola A., Fontana D., and Gambacorti-Passerini C., Molecular Cancer Research 2014; Cecon M. et al., Mol Cancer Res. 2013 Feb;11(2):122-32. doi: 10.1158/1541-7786.MCR-12-0569; Debruyne DN et al., Oncogene, 2016 35:3681-3691). Here the authors show that IC50 levels are around 500 nM, which seems to be rather high, which should be explained/investigated deeper. Further, the authors employ brigatinib/ceritinib but show no immunoblots for phosphorylated ALK or if ALK is expressed, for each dose used in the assay, which should be included.
5. The authors also perform the overexpression screen in the CHLA-20 cell line. They only put the raw data in the supplementary table 10. They need to provide the analyzed CHLA-20 data.
6. The authors only examined the overexpression of five genes by RT-qPCR (Supplementary figure 2C). They should investigate the expression level of all 25 genes which were tested in the dose-response assays.
7. In the Kaplan-Meier analysis, firstly, the authors show that high level of PIM1 has a better prognosis than low expression of PIM1, something is wrong here. The authors used the optimal threshold for each gene. On the other hand, these Kaplan-Meier analysis of PIM1 has already been shown by Brunen D., de Vries RC., Lieftink, C., Beijersbergen R., and Bernards R., Molecular Cancer Therapeutics, 2017, although they show that high level of PIM1 has worse prognosis. Maybe the authors can refer to their article instead, since it already been published. If the authors compare PIM1 prognosis with a high-risk neuroblastoma gene, such as MYCN, is there a difference of prognosis?
8. They should provide quantified results of Fig2D. It is rather clear that BAD expression level is higher in the gRNA treated lanes. They author claims this is important, therefore they should calculate and discuss the ratio between pBAD/BAD.
9. There are type errors in the main text.
10. In main text. Supplementary Figure "SC" should be "3C".
In main text. ...overexpression of PIM1 (Figure 2C) should be "Supplementary Figure 4C".
11. In main text....exceeding 10uM in 9/9 NB cell lines should be 7/7
12. Figure 4.
In SH-SY5Y. AZD1208 100nM and brigatinib 100nM. CI=1.09. Remove bold/italic

In CHLA-20. AZD1208 60nM and brigatinib 30nM. CI=0.81. Add bold/italic

13. PIM1 knockdown efficiency is not good enough in CHLA-20 Fig3E and sensitive to drugs may due to off-target effect. PIM1 knockdown should be tested in a similar manner in the other cell line too, SH-SY5Y.

14. In figure 4, the synergistic effect of AZD1208 and brigatinib or AZD1208 and ceritinib may not be significant as mentioned in the main text.

15. AZD1208 and brigatinib. 35 out of 49 were CI>1.

AZD1208 and ceritinib. Only 7 out of 49 passed the author's criteria (CI<=0.95).

27 out of 49 were CI>1 and rest of them were $0.9 < CI < 1$.

16. In figure 4, The authors should provide information in the legend indicating the meaning of each colored square and the number in the square should be labeled correctly with bold face. Is the color indicating levels for combinations treatment in xenografts or in human?

17. Figure 4B, what is the fixed concentration in nM employed for brigatinib treatment?

18. Figure 5B, Xenograft study, when did treatment of the mice start? In the materials and method it is mentioned that "once tumours reached 100-200 mm³, mice were randomly split into four treatment groups". In 5B is clear that treatment of the xenografts start below the level stated in material and methods. This should be changed in the figure or changed in material and methods.

19. Sentence, "single-agent treatment with ceritinib of AZD1208 led to a significant delay in tumor growth relative to vehicle treatment". Maybe the authors should mention or emphasize that the delay was not long term and combination treatment lasted only for roughly 22 days.

20. Fig 5E mislabeled y-axis, It should be normalized to the vehicle.

21. Wording, "consistent with the superior efficacy of combination treatment", this study is preclinical and superior is a more correct word in clinical trials.

22. The authors should mention and discuss the articles Debruyne DN., et al., Oncogene (ref 28 in your manuscript) and Wang, HQ., et al., eLIFE Elife. 2017 Apr 20;6. pii: e17137. doi:

10.7554/eLife.17137, which deal with resistance in human ALK mutant neuroblastoma cell lines.

Reviewers' Comments:

Reviewer #1:

Remarks to the Author:

The authors made extensive revisions that addressed the most important aspects of the previous critiques. While I do think that the manuscript would benefit from assessment of cell death in the in vivo models, the in vitro results are convincing.

I recommend removing the Cleaved Caspase 3 staining from Supp Fig 8f and corresponding text, as it will likely cause confusion.

I am happy to recommend this article for publication

Reviewer #2:

Remarks to the Author:

The authors Trigg and colleagues present their revised manuscript which examines the role of PIM1 inhibition in ALK-expressing neuroblastoma. With regards to this reviewer's concerns, the authors have addressed each of the previous points. Most notably, the authors have tested additional cell lines for both MYCN amplified and non-amplified neuroblastoma, and have found that PIM1 inhibition enhances the effect of ALK inhibition in ALK expressing neuroblastoma. The authors also have addressed the statistical issue by performing Mann-Whitney statistical analysis in their animal studies, and they still find significance.

With the revisions and additional data, the manuscript has become quite unwieldy and difficult to follow and read. In particular, the first results section extends from page 4 to 6, contains one figure, 5 supplementary figures, and 5 supplementary data sets. This section which could be divided into several sections is too long and incorporates excessive detail on methods. It is challenging to follow as the results bounce around from supplementary figure to figure 1 to supplementary datasets. Very difficult to follow and hence understand. As was an issue with the first version of the manuscript, figures do not follow in numeric order - as Supplementary Figure 4 comes before Supplementary Figure 3.

Reviewer #3:

Remarks to the Author:

The major/massive concern regarding this manuscript is that the synergy of ALK and PIM1 inhibitors is close to non-existing (Figure 4), which the authors also have realized by now and change the head line of the manuscript from synergy to "PIM1 inhibition enhances the sensitivity of high-risk aberrant ALK-expressing neuroblastoma to ALK inhibition regardless of MYCN status". The synergy scope is missing. Secondly, PIM1 kinase has already been suggested as a target for neuroblastoma employing a CRISPR-Cas9 screen which identified genes whose loss confers PIM inhibitor resistance (Brunen D., de Vries RC., Lieftink, C., Beijersbergen R., and Bernards R., *Molecular Cancer Therapeutics*, 2017.) Similar to Trigg they observed that AZD1208 initially suppressed growth of Kelly neuroblastoma ALK-F1174L xenografts but did not prevent long term tumor growth. Further, Debryne, DN et al., *Oncogene*. 2016 Jul 14;35(28):3681-91. doi: 10.1038/onc.2015.434., show that ALK inhibitor resistance in ALK-F1174L driven neuroblastoma is associated with AXL activation and induction of EMT not found by Trigg and coworkers.

**Response to reviewer's comments, Trigg et al., PIM1 inhibition enhances the sensitivity of high-risk aberrant ALK-expressing neuroblastoma to ALK inhibition regardless of MYCN status
NCOMMS-19-03936A**

Many thanks to the reviewers for taking the time to assess this substantial body of work. Please find our responses to reviewers' comments below in blue:

Reviewer #1:

The authors made extensive revisions that addressed the most important aspects of the previous critiques. While I do think that the manuscript would benefit from assessment of cell death in the in vivo models, the in vitro results are convincing. I recommend removing the Cleaved Caspase 3 staining from Supp Fig 8f and corresponding text, as it will likely cause confusion. I am happy to recommend this article for publication

Many thanks to the reviewer for considering the additional data and considerable extra information we have provided in the revised manuscript. We have now removed Supplementary Fig. 8f, and all mentions of it from the text (page 9, lines 274-280) and from Supplementary Table 5.

Reviewer #2:

The authors Trigg and colleagues present their revised manuscript which examines the role of PIM1 inhibition in ALK-expressing neuroblastoma. With regards to this reviewer's concerns, the authors have addressed each of the previous points. Most notably, the authors have tested additional cell lines for both MYCN amplified and non-amplified neuroblastoma, and have found that PIM1 inhibition enhances the effect of ALK inhibition in ALK expressing neuroblastoma. The authors also have addressed the statistical issue by performing Mann-Whitney statistical analysis in their animal studies, and they still find significance. With the revisions and additional data, the manuscript has become quite unwieldy and difficult to follow and read. In particular, the first results section extends from page 4 to 6, contains one figure, 5 supplementary figures, and 5 supplementary data sets. This section which could be divided into several sections is too long and incorporates excessive detail on methods. It is challenging to follow as the results bounce around from supplementary figure to figure 1 to supplementary datasets. Very difficult to follow and hence understand.

We agree that the additions and all of the supplementary data have rendered the manuscript difficult to read. To remedy this issue, we have introduced sub-sections to the results with new titles to draw out the significance of these data. We have also deleted some of the methodology but have left some in place as this is a significant amount of work and the validation steps of the CRISPR screen are an important undertaking that should not be underestimated, particularly as they speak towards the robustness of our data. Specifically, these data are exceptionally important to readers of the manuscript as the methodology used is relatively novel and CRISPR screen results can be highly dependent on the methodology. As such, we hope by leaving these data in place, the readers of the manuscript will be able to recapitulate our data and understand fully how we arrived at our conclusions. However, if the reviewer feels that this is still too much information, we

can attempt to remove more. Regardless, we hope that these smaller blocks of results now make the manuscript more accessible and clarify the key messages of our work.

As was an issue with the first version of the manuscript, figures do not follow in numeric order - as Supplementary Figure 4 comes before Supplementary Figure 3.

We have made certain that all figures follow in order. This has led to some rearrangement and renumbering of the figures as indicated in the text.

Reviewer #3:

The major/massive concern regarding this manuscript is that the synergy of ALK and PIM1 inhibitors is close to non-existing (Figure 4), which the authors also have realized by now and change the head line of the manuscript from synergy to "PIM1 inhibition enhances the sensitivity of high-risk aberrant ALK-expressing neuroblastoma to ALK inhibition regardless of MYCN status". The synergy scope is missing.

Whilst the synergy between the drugs is mild in cases *in vitro*, it is still existent. We refer the reviewer specifically to page 8, lines 244-255 in particular whereby synergy is seen *in vitro*, albeit relatively mild. Perhaps more important in this regard is the *in vivo* data we present with patient derived xenografts (rather than cell line xenografts) - perhaps more clinically applicable and is clearly significant with mouse survival considerably increased. The change in title of the manuscript is not to reflect a lack of synergy, but rather the perhaps more important observation that this therapeutic approach is relevant regardless of the MYCN status of the tumour. To address the reviewer's concerns regarding the mild *in vitro* synergy, we have added to the discussion of the paper to emphasise the more relevant *in vivo* findings (page 12, lines 374-378 and lines 378-388):

"To examine the clinical relevance of this drug combination, we employed two patient-derived models of high-risk NB harboring ALK^{F1245C} or ALK^{F1174L} mutations respectively. PDX are perhaps more representative of the clinical scenario than are cell lines due to their relative genetic likeness to tumours in patients, as opposed to long-term cultured cell lines that have evolved further from the original malignancy. However, our data, particularly the *in vivo* PDX data suggest that combined PIM and ALK inhibition is effective regardless of MYCN status."

Secondly, PIM1 kinase has already been suggested as a target for neuroblastoma employing a CRISPR-Cas9 screen which identified genes whose loss confers PIM inhibitor resistance (Brunen D., de Vries RC., Liefink, C., Beijersbergen R., and Bernards R., Molecular Cancer Therapeutics, 2017.) Similar to Trigg they observed that AZD1208 initially suppressed growth of Kelly neuroblastoma ALK-F1174L xenografts but did not prevent long term tumor growth. Further, Debryne, DN et al., Oncogene. 2016 Jul 14;35(28):3681-91. doi: 10.1038/onc.2015.434., show that ALK inhibitor resistance ALK-F1174L driven neuroblastoma is associated with AXL activation and induction of EMT not found by Trigg and coworkers.

We respect the articles cited by the reviewer and had included references to these in our manuscript to acknowledge the significant work conducted by the authors of those papers (page 6, line 172; page 6, line 179; page 7, line 224; page 10, line 331-334). Together, with these manuscripts, our data provide important information towards the therapeutic management of

patients with ALK-expressing neuroblastoma that add significantly to the literature. In particular, unlike the paper by Brunen et al., we employ a CRISPR overexpression screen (as opposed to a knockout screen used in the Brunen paper) and furthermore employ it to understand resistance mechanisms to ALK TKIs rather than PIM inhibitors. Indeed, one might take from both our own and their sets of data that combined inhibition of ALK and PIM upfront (for ALK expressing NB) might prevent resistance to both ALK and PIM kinases from developing, although this of course requires validation. As such, we feel our data are no way in conflict with those presented in this publication, but rather support its findings and add additional important information towards the treatment of children with ALK-expressing NB.

Reviewers' Comments:

Reviewer #2:

Remarks to the Author:

The rework and reorganization of the manuscript is much improved and better structured.

The authors have addressed my critiques and I would favor publication.

REVIEWERS' COMMENTS:

Reviewer #2 (Remarks to the Author):

The rework and reorganization of the manuscript is much improved and better structured. The authors have addressed my critiques and I would favor publication.

We thanks reviewer 2 for their approval of the manuscript